# Identifying Plant Physiological and Climatic Drivers in the Woody Shrub *Prosopis strombulifera*: Effects of Spring Traits on Flower Sprouting and Fruit Production

Felipe S. Carevic [1,2,*], Roberto Contreras [2,3], Erico Carmona [4] and Ermindo Barrientos [5]

1   Laboratorio de Ecología Vegetal, Facultad de Recursos Naturales Renovables, Universidad Arturo Prat, Campus Huayquique, Iquique 1100000, Chile
2   Millennium Nucleus on Applied Historical Ecology for Arid Forests [Aforest], Santiago 8320000, Chile
3   Centro Regional de Investigación y Desarrollo Sustentable de Atacama [CRIDESAT], Universidad de Atacama, Copayapu 485, Copiapó 1530000, Chile
4   Laboratorio de Bionanomateriales, Facultad de Recursos Naturales Renovables, Universidad Arturo Prat, Campus Huayquique, Iquique 1100000, Chile
5   Engineering Agriculture Department, Natural Sciences and Agriculture Faculty, Universidad Técnica de Oruro, Oruro 00591, Bolivia
*   Correspondence: fcarevic@unap.cl

**Abstract:** *Prosopis strombulifera* is a widely distributed woody species distributed along arid ecosystems in America. The interannual evolution of ecophysiological parameters and their effects on fruit production and flower sprouting in *Prosopis strombulifera* were studied for three years in a natural population distributed in the Atacama Desert. Xylem water column tension, pressure–volume curves, specific leaf area (SLA), and chlorophyll fluorescence parameters were assessed. Flower sprouting was assessed in different weeks using tagged flowers. To assess fruit production, four small containers were placed under twenty-five individuals, allowing the estimation of total annual production and individual production. We found considerable variability between years and between individuals. Positive relationships were found between plant water parameters, SLA, and chlorophyll variables measured in spring at flower sprouting and during fruit production. A negative correlation was found between the mean of the minimum temperatures in spring and flower sprouting. These results suggest that spring ecophysiological parameters strongly affect the reproductive status of *P. strombulifera*. The results also reflect the potential of this species to adapt to a hyperarid climate by preserving a high relative water content before flower sprouting.

**Keywords:** hyperarid climate; Atacama Desert; *Prosopis*; flower sprouting; ecophysiology

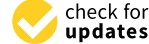



## 1. Introduction

Habitually, fruit production plays an instrumental role in the formation and equilibrium of forest systems in deserts and other areas prone to water stress, as reproduction parameters play important ecological roles in maintaining natural regeneration. In deserts and other areas prone to drought, the regeneration of woody species from seeds is more important than asexual reproduction or clonal growth in maintaining stable populations [1]. Nevertheless, the ecophysiological traits and climatic drivers determining both flower sprout and seed output remain poorly understood in woody species of arid ecosystems [2].

Desert plants exhibit high seasonal physiological plasticity and variability at the intra- and inter-population levels, which likely influence reproductive behavior [2,3]. In this context, one of the most widely distributed tree genera in arid ecosystems is *Prosopis*, whose fruit production dynamics are strongly influenced by climatic parameters. For example, studies on *Prosopis* tree species demonstrated the effects of the mean minimum temperatures of winter and spring xylem water potentials on flower production during the summer [4]. However, the influence of seasonal ecophysiological parameters and their effects on fruit

production in *Prosopis* species growing in hyperarid conditions are poorly understood. It is important to consider plant–water relations, and climate may play important roles in monitoring and predicting plant viability [5]. According to Pesendorfer et al. [5], the main factors that regulate reproductive traits such as flower production and fruit production are directly related to climatic factors, which determine the physiological behavior of species that present some degree of variability in seed production, a phenomenon known as "masting".

Fruit production is a key factor concerning the dynamics of plant ecosystems because it influences the regeneration of species and plays an essential role in feeding birds and small mammals associated with natural ecosystems [6]. In this context, successful reproduction parameters in plants is a complex process driven by the interplay of multiple biotic and abiotic factors impinging on the different life cycle stages of plants. The main requisite involves the existence of a suitable pollen generation and seed supply depending on stand maturity, climate, fertility, and masting [5,6]. Furthermore, fruits constitute a part of the diet of domesticated camelids, goats, and sheep. In the ecosystems of northern Chile, mesquite (*Prosopis*) species are particularly important. These trees contribute to the ecological and socioeconomic stability of agriculture by native farmers in natural areas termed "pampas". These regions are characterized by low plant and animal biodiversity [7]. *Prosopis strombulifera* is an insect-pollinated halophyte that is common in Patagonia and central Argentina, northern Chile, and central Peru in South America, as well as in Arizona and the Imperial Valley in California, USA. In some areas, such as the desert rangelands, this species is considered invasive and poses a threat to native grassland systems [8,9]. Conversely, in northern Chile, this woody phreatophyte species is used as a food source for domestic cattle. Additionally, its natural populations provide habitats for fauna, serve as carbon sinks, and enrich the soil through nitrogen fixation [10]. The main threat to natural *P. strombulifera* populations in northern Chile is associated with the current depletion of the water table, which may be a direct cause of population declines [11]. The local water table is fed by summer precipitation in the Andean mountains in northern Chile [12]. Thus, groundwater aquifers are the only source of water, which are fed by rainfall, glaciers, and snowmelt from the Andes. We hypothesized that seasonal ecophysiological parameters would strongly affect reproduction parameters. In this regard, the main objective of this study was to determine the effects of seasonal plant ecophysiological parameters on reproductive traits (i.e., flower sprouting and fruit production) in the woody shrub *Prosopis strombulifera* growing in the hyperarid Atacama Desert.

## 2. Materials and Methods

### 2.1. Study Area

This study was carried out within the natural habitat of *Prosopis strombulifera* in Pampa del Tamarugal, northern Chile. During the period 2015–2018, an experimental grid of 3.8 ha was established in the La Huayca sector in the Pampa del Tamarugal, northern Chile ($20°24'42''$; $69°36'58''$; 985 m above sea level) to evaluate plant physiological parameters in a *P. strombulifera* population. Density was estimated at 24 ind/ha$^{-1}$. The climate in this area is hyperarid. Rainfall, averaging 0.6 mm per year, is very scarce, and the average annual temperature is 20.9 °C [13]. The vegetation of the area is dominated by woody mesquite species (*Prosopis strombulifera*, *Prosopis tamarugo*, *Prosopis alba*, and *Prosopis burkartii*), with an understory predominated by *Distichlis spicata*. For environmental characterization, one meteorological station was established near the study area to record daily environmental conditions (ambient temperature, precipitation, wind speed, and solar radiation) and was located 1.5 km from the study area ($20°26'39''$; $69°32'07''$; 1005 m above sea level). Daily precipitation and temperature data are summarized in Figures 1 and 2.

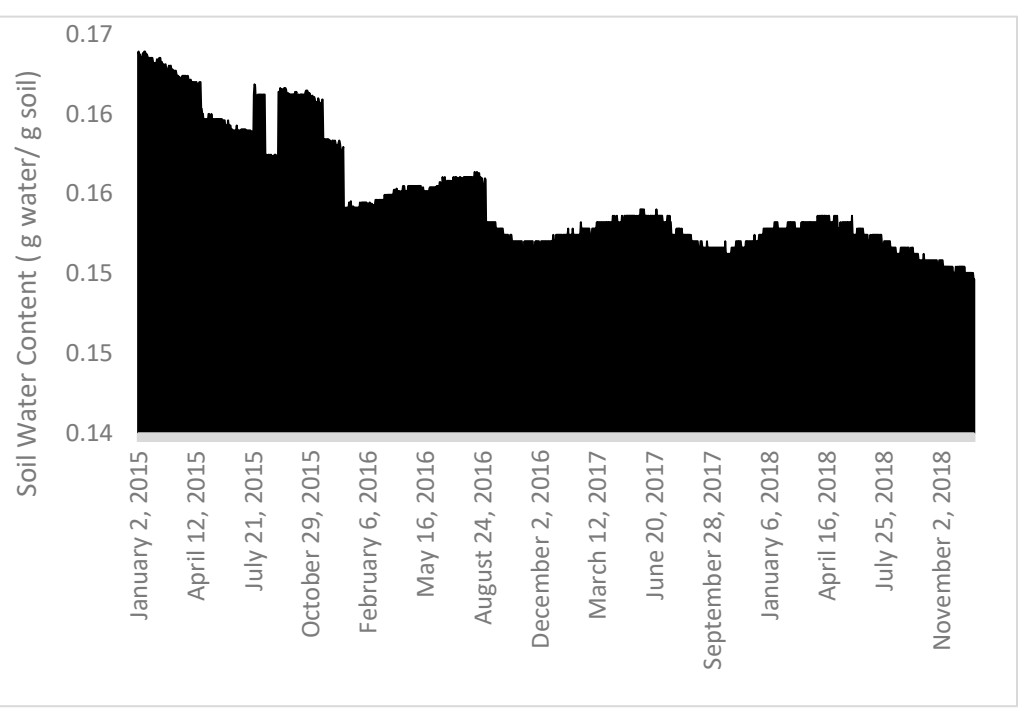

**Figure 1.** Soil wetness determined during the entire study.

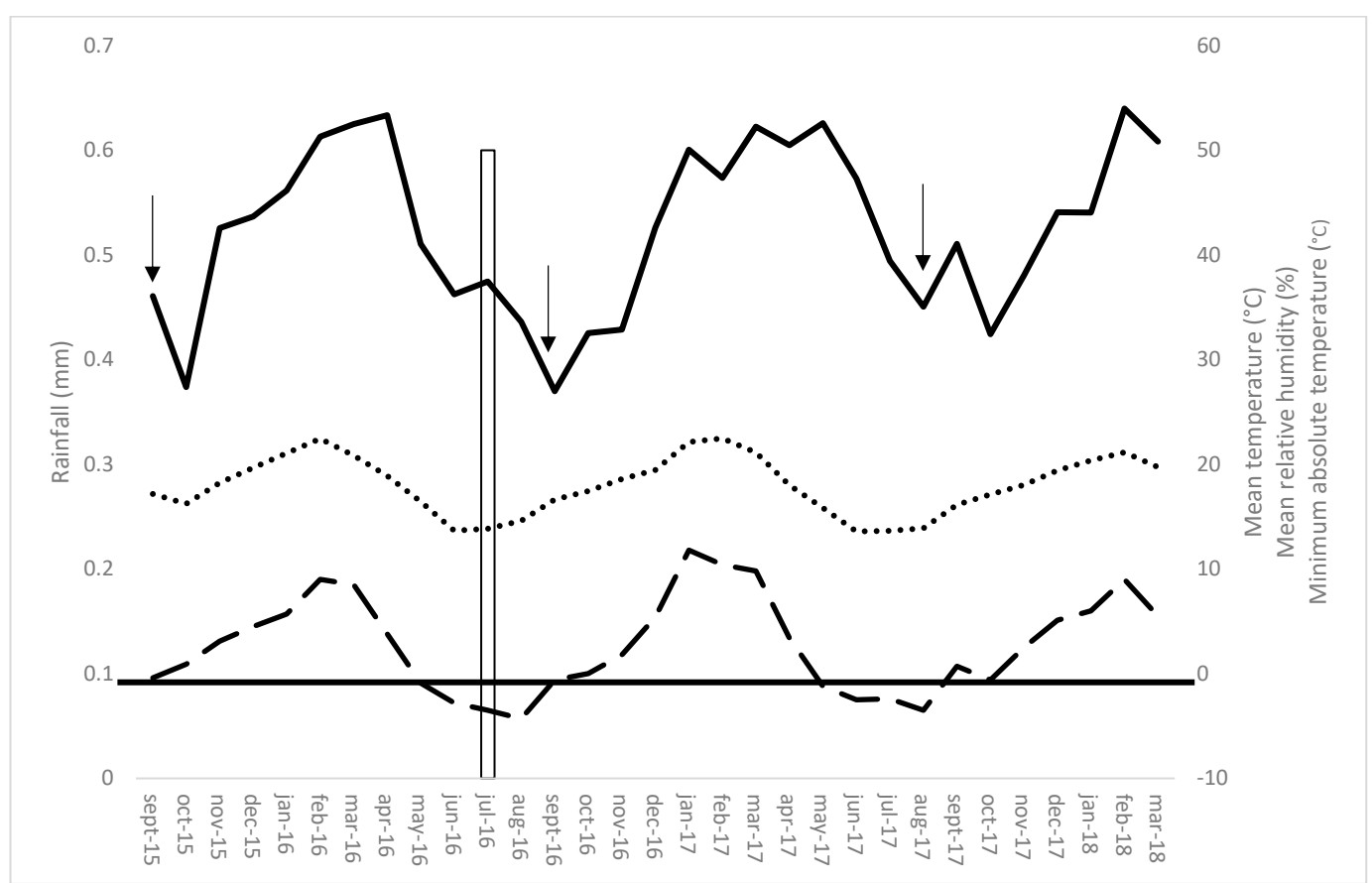

**Figure 2.** Meteorological data obtained by the weather station: rainfall (white bar); mean monthly temperature (dotted line); minimum absolute temperature registered (dashed line); and mean relative humidity (black line). Arrows indicate the beginning of the reproduction period of *P. strombulifera*. The straight black line denotes a temperature of zero degrees Celsius.

### 2.2. Water Parameters and Leaf Traits

Water parameters and leaf traits were measured seasonally over three consecutive years, from December 2015 to May 2018. A total of 25 individuals were chosen randomly for all ecophysiological measurements. Digital humidity probes (ECH$_2$O EC-5 Sensor) and a 12-bit A/D (Decagon, WA, USA) were mounted next to each individual (at 1.2–1.7 m distance) to monitor soil water content at 90 cm depth (Figure 1). The shoot water potential ($\Psi$) at pre-dawn was measured in situ using a pressure chamber (1505D; PMS Instruments, Corvallis, OR, USA). On each sampling date, two shoots per shrub were cut from the external part of the crown at each shrub's medium height in eastward and westward directions, between 06:00 and 07:00 h. At the same time, two more shoots per shrub were cut, which were kept refrigerated and in the dark with their basal ends immersed in distilled water until they were transferred to the laboratory. We had checked in advance that there were no significant differences between the measurements taken in the field immediately after they were cut and those taken in the laboratory after their transfer. These shoots were subsequently used to produce isothermal pressure–volume curves. *P. strombulifera* height was measured using a Nikon™ Forestry Pro laser hypsometer. The laser hypsometer provides a three-point measurement mode that calculates the horizontal distance to the shrub, then measures the angles to the top and base to calculate the height between the two points.

To construct pressure–volume curves, ten shoots were collected per season. These shoots were cut from larger samples that had been placed in a dark 12-V refrigeration chamber (3 °C) with their bases immersed in distilled water. Additionally, they were covered with a polyethylene bag for 24 h to improve hydration. Further details on the pressure–volume curve methodology were described by Corcuera et al. [14]. Data were obtained using the free transpiration method [15], which consists of recording $\Psi$ and fresh weight of the shoots over short periods under a constant temperature of $25 \pm 2$ °C until dehydrated. The shoots were then placed in an oven at 70 °C to determine their dry weight and relative water content (RWC). After plotting the pressure–volume curves, as no over-saturation points were detected on any of them, the following parameters were determined: osmotic potential at full turgor ($\Psi\pi_{100}$), osmotic potential at the turgor loss point ($\Psi\pi_0$), relative water content at the turgor loss point (RWC$_0$), and apoplastic relative water content (RWC$_a$). This last parameter plays an important role as an indicator of the water content in extracellular spaces in cells. The specific leaf area (SLA) was determined by calculating the ratio of leaf area to leaf dry weight (g/cm$^2$). Leaf area was determined using a leaf area meter (LICOR 3000C), and leaf dry weight was recorded after drying at 70 °C for two days.

### 2.3. Photosynthetic Efficiency

Photosynthetic efficiency was measured using a portable fluorometer (MultispeQ Instrument, Michigan State University, MI, USA). The data were analyzed using the PhotosynQ web portal www.photosynq.org (accessed on 3 May 2023), which allowed the rapid estimation of chlorophyll fluorescence. In this context, MultispeQ combines a pulse-amplitude-modulated fluorometer and a spectrometer into one . Three leaves per individual were collected in each season and kept in the dark with a clip for 5 min to allow all reaction centers to open and to minimize fluorescence, according to Guisande et al. [16]. In addition, the following parameters were determined: ratio of variable fluorescence to maximal fluorescence (Fv/Fm), nonphotochemical quenching (NPQ), and photochemical quenching (Pq). Chlorophyll content was estimated using the SPAD index with a hand-held chlorophyll content meter (SPAD-502, Minolta, Ramsey, NJ, USA). For all experiments, photosynthetic and chlorophyll content measurements were taken on leaves that were fully expanded and mature.

### 2.4. Flower Sprouting and Pod Tagging

Flower sprouting was assessed in the same 25 individuals selected for the other measurements in different weeks during the austral spring, from November to January, during the period of 2015–2016 to 2017–2018. To track the emergence of flowers throughout each week, a specific colored thread was used to tag the flowers. Each week had a different thread color assigned to it, allowing for the identification and allocation of flower production and pods to their respective weeks of sprouting at the time of harvest. Thus, the start and end of flowering and flower sprouting were documented [17]. This tagging was also used to record the total number of flowers produced per plant (total number of tags per plant), the number of pods per plant (number of tags with a date of a flower sprouting), the number of fertile pods (number of fruits per pod and plant, counted at harvest), and the proportion of aborted flowers and pods.

### 2.5. Fruit Production

Fruit production was quantified using the fruit container method of Greenberg [18] during three fruiting periods, i.e., during the saustral summer from January to April 2016–2018. Four small circular containers of 0.15 m in diameter were placed beneath the shrubs. The containers were placed under the crowns of the individuals and faced northwards, southwards, eastwards, and westwards, at a distance of $\frac{3}{4}$ of the crown radius in each direction. Samples were collected weekly from the 25 individuals used for measuring ecophysiological parameters. The fruits were transported to the laboratory to be measured for mass using a precision scale. We calculated fruit mass as fresh mass per m$^2$ of shrub crown projection area (g/m$^{-2}$). We used these data to compare fruit production between years and between individuals. Fruit moisture was calculated according to the following equation: FM = ((FW − DW)/FW) × 100, where FM is fruit moisture and FW and DW are fresh and dry weights, respectively. Fresh and dry weights were recorded using an analytical balance (MS204TS; Mettler Toledo, Greifensee, Switzerland). Dry weight was recorded after drying in an oven (UN110; Memmert, Schwabach, Germany) at 80 °C for 72 h. Fruit size was estimated by determining length using a manual caliper (799A; Starrett, Suzhou, China) and was expressed in millimeters.

### 2.6. Data Analyses

Differences in fruit production rates between years and between individuals were tested with a repeated-measures ANOVA using "year" (2016–2018) and "individual" as factors, followed by Tukey's honest significance test. Normality and equality of variances were tested using the Kolmogorov–Smirnov test. The coefficient of variation (standard deviation/mean) among individuals was calculated for each year to assess the mean annual fruit production and flower sprouts. Relationships of ecophysiological parameters and climate with fruit and flower production were estimated using Spearman correlation analyses integrating the set of 75 observations carried out during the study. Following the regression analysis, the Durbin–Watson coefficient (DWC) was used to determine the autocorrelation of the residuals. The presence of correlated residuals indicates a violation of an important assumption of a least-squares regression. All statistical analyses were performed using SPSS 16.0 (IBM, Armonk, NY, USA). Criteria to perform and interpret statistics were applied according to Sokal and Rohlf [19].

## 3. Results

Ecophysiological parameters differed between seasons, and SLA, NPQ, Fv/Fm, and RWC$_a$ showed marked trends throughout the study (Figure 3).

The average individual height was 1.38 ± 0.11 m. Soil moisture was not correlated with reproductive parameters ($p$ = 0.15; F = 198.003). Fruit production differed significantly between years ($p$ = 0.03; F = 11.98), with a maximum of 178.1 ± 3.4 g/m$^{-2}$ in 2018 and a minimum of 142.9 ± 5.8 g/m$^{-2}$ (mean ± SE) in 2017. We observed significant differences in fruit production between individuals in the years 2017 and 2018 (Table 1; intrapopulation

data), with values ranging from 55.78 to 156.02 g/m$^{-2}$ (fresh weight), as observed in 2017 and 2018, respectively. The overall average fresh weight per fruit was $5.88 \pm 0.8$ g (n = 144). Regarding flower sprouts, significant differences were observed only in 2017 (the year with the lowest fruit production), and flower production differed significantly between individuals (Table 1).

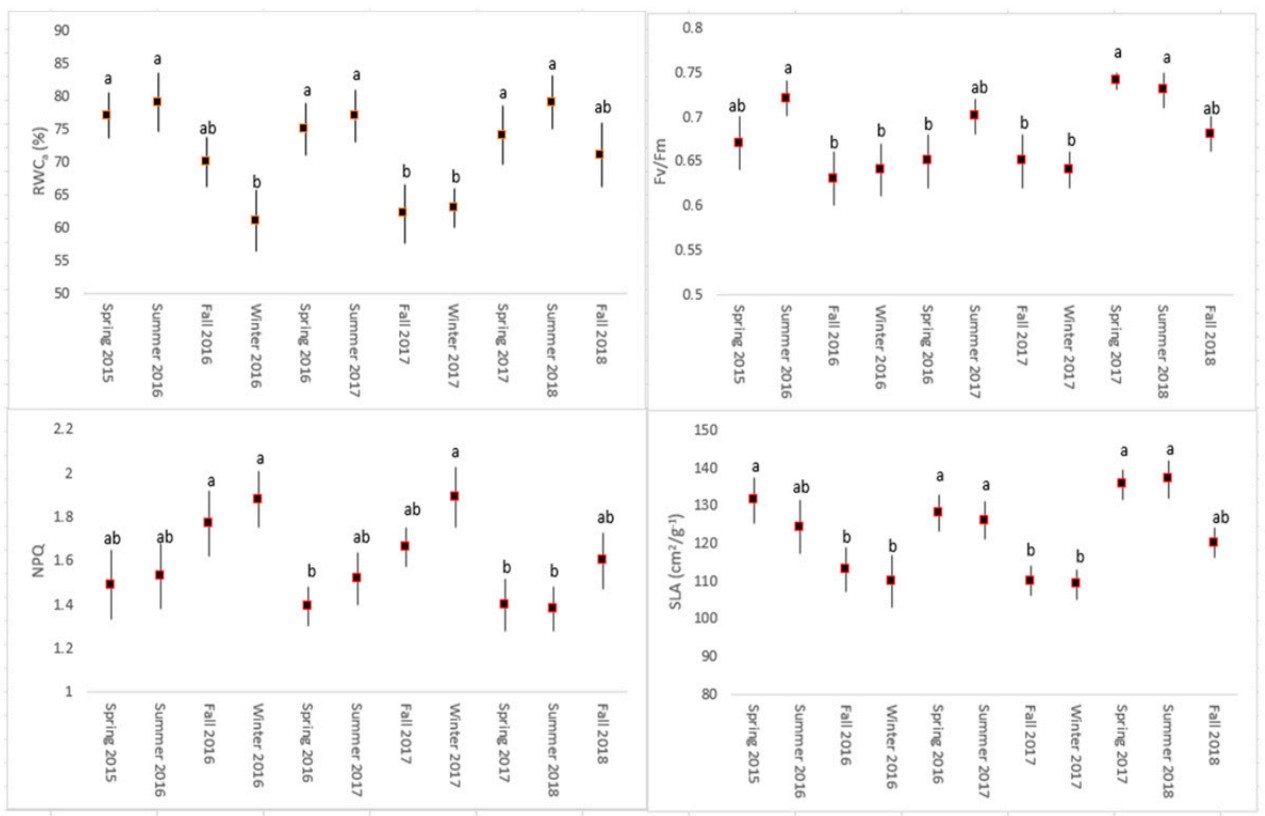

**Figure 3.** Seasonal variation of parameters of RWC$_a$, apoplastic relative water content, Fv/Fm, ratio of variable fluorescence to maximal fluorescence, NPQ, nonphotochemical quenching, and SLA, specific leaf area. Significant differences between dates are denoted by letters ($p < 0.05$).

**Table 1.** Summarized data on fruit production and flower sprouts at the intrapopulation and interannual levels. Mean $\pm$ SE. Significant differences between years are denoted by letters ($p < 0.05$).

| | Fruit Production | | | | Flower Sprout | | | |
|---|---|---|---|---|---|---|---|---|
| | Intrapopulation | | | Interannual | Intrapopulation | | | Interannual |
| Year | *p*-Value | F | Variation Coefficient | Total Fruit Biomass [g m$^{-2}$] | *p*-Value | F | Variation Coefficient | Mean Number of Flower Sprouts |
| 2016 | 0.23 | 0.18 | 0.65 | 169.8 [a] $\pm$ 6.1 | 0.12 | 1.08 | 0.55 | 26 [a] $\pm$ 5.3 |
| 2017 | 0.03 | 0.11 | 0.74 | 142.9 [b] $\pm$ 5.8 | 0.04 | 1.87 | 0.58 | 12 [b] $\pm$ 4.1 |
| 2018 | 0.22 | 0.19 | 0.75 | 178.1 [a] $\pm$ 3.4 | 0.11 | 2.99 | 0.60 | 24 [a] $\pm$ 5.2 |

Flower buds were positively correlated with fruit production (r = 0.89; $p < 0.01$; F = 102.873). The ecophysiological values that presented a close relationship with the production of flowers and fruits were the values obtained during the spring season of each year (Table 2). Thus, the parameters that directly influenced these traits—at a significant level—were RWC$_a$, SLA, Fv/Fm, and NPQ. Regarding climate, mean humidity in September was positively correlated with flower production (r = 0.53) and with the mean minimum temperature recorded in October (r = $-0.61$; $p = 0.02$, F = 88.987).



**Table 2.** Plant ecophysiological parameters obtained during the spring season in *Prosopis strombulifera* and their relationship with fruit production and flower sprouts were obtained by correlation analysis. Significant relationships are denoted by bold type numbers (<0.05). r: Spearman correlation coefficients; *p*-value: statistical significance.

| Parameter [n = 75] | Fruit Production | | Flower Sprout | |
|---|---|---|---|---|
| | *p* Value | r | *p* Value | r |
| $\Psi_{predawn}$ | 0.43 | <0.01 | 0.54 | <0.01 |
| $\Psi\pi_{100}$ | 0.93 | <0.001 | 0.41 | <0.01 |
| $\Psi\pi_0$ | 0.13 | 0.10 | 0.29 | 0.09 |
| $RWC_0$ | 0.19 | 0.17 | 0.45 | 0.02 |
| $RWC_a$ | 0.02 | 0.55 | >0.01 | 0.58 |
| Fv/Fm | 0.09 | 0.32 | 0.09 | 0.29 |
| NPQ | 0.03 | −0.52 | 0.01 | −0.50 |
| Pq | 0.55 | <0.01 | 0.36 | <0.01 |
| SPAD | 0.88 | <0.001 | 0.43 | <0.01 |
| SLA | 0.04 | 0.48 | 0.02 | 0.45 |

## 4. Discussion

This is the first study to assess the effects of climate and ecophysiological variables on the reproductive parameters of *P. strombulifera*. The observed positive correlation of spring values of $RWC_a$ with flower and fruit production was as expected because this behavior is strongly associated with the passive concentration of solutes to maintain turgor during the reproductive stage [20]. RWCa is closely linked to water flow in *Prosopis*, specifically to hydraulic conductivity parameters [*K*] [21]. During the winter months, this flow decreases significantly due to cold cavitation. This process reduces the apoplastic water content. However, as spring and summer arrive, these hydraulic flows are restored. This restoration coincides with an increase in temperature and the recharge of the water table, which increases water availability for *Prosopis* plants in Pampa Tamarugal [12,22].

Xylem water variables are one of the most important physiological parameters for plants living in hyper-arid zones, as they provide insights into their adaptations to extreme conditions typical of zones under water stress. For example, seasonal analyses of plant water efficiency in xeric ecosystems are useful to investigate the role of abiotic parameters, such as winter frosts and summer droughts [23]. Unlike desert plants in the northern hemisphere, strategies of adaptation for desert plants in the Atacama Desert show rapid growth and reproduction in short periods when water is available [24]. Thus, the measurement of variables derived from plant water and osmotic potentials under natural conditions of growth in desert plants seems to be a useful strategy to increase knowledge of growth traits under natural conditions [25]. In this context, Bartlett et al. [26], have established the importance of leaf water potential at turgor loss, elasticity modulus ($\epsilon$), and apoplastic relative water content ($RWC_a$) as traits predictive of drought tolerance at biome scales, where this last parameter seems to be the most sensitive parameter for predicting the distribution of species in arid zones. In our study, $RWC_a$ was strongly correlated with fruit production and flower parameters, which is indicative of the high dependence of this species on water transport in cell walls during the pre-reproductive period. This information could be valuable to establish potential risks for environmental health and, at the moment, identify the lowest values of plant water relations, specifically $RWC_a$. This has led many authors to suggest that physiological parameters could be involved in the most important productive process in plants: seed production [4]. Despite its importance, some preliminary studies on arid species are in order regarding measuring the effect of induced water stress on photosynthesis and net $CO_2$ uptake without considering the effect of environmental variables on reproductive processes [27]. In this context, we found a negative relationship between the minimum temperatures detected in spring and the number of flower sprouts, transforming a vegetative meristem into a flowering meristem. Similar

results were found in species of this same genus [4,22], which undoubtedly reflects that cold temperatures at this time of year are a limiting factor for flowering.

Individuals of *P. strombulifera* exhibited marked leaf plasticity in spring (before flower formation) and partly during the summer, which was associated with the production of wider leaves with larger leaf mass through increased SLA. In contrast, during the coldest periods of the year, SLA decreased notably to its minimum. In this sense, the most likely interpretation of this fluctuation refers to a typical trait of adaptation to arid habitats of the genus *Prosopis*: during periods of higher environmental stress, leaves tend to be thicker to avoid the effects of frost [4]. By contrast, when temperatures and photoperiods increase, leaves tend to be wider and thinner to increase photosynthetic activity and compensate for higher requirements during the period of flower production [28,29]. The highest NPQ value was detected during the winter, which was directly related to higher efficiency in the heat dissipation process. Hamerlynck and Huxman [30] described this response as a photoprotective mechanism of desert species to prepare for reproduction. In this sense, high NPQ during the cold season may be a measure of photoinhibition due to low temperatures and shorter photoperiods. This trend was also reflected in the respective Fv/Fm values. The Fv/Fm ratio represents the maximum efficiency of photosystem II, where a low Fv/Fm value indicates inefficient use of the absorbed energy and suggests photoinhibition [31], which in the present study was observed during the cold season. In spring, when conditions are more favorable, NPQ is lower, and significantly more light is diverted to photochemistry in order to support growth [32].

Soil humidity ranged from 20% to 23%, with no difference between years. These ranges were in line with those of Aravena and Acevedo [33] and McKay et al. [34]. This soil moisture recorded in the first centimeters of depth originates from the underground aquifer, is established at this level due to capillary formations, and would be of use for vegetation in periods during which the water table decreases [35]. The lack of correlation between topsoil moisture and reproductive variables in this species is, however, not surprising, as these results in fact confirm the phreatophytic habit of *P. strombulifera* in this ecosystem. Similar results were obtained in *Prosopis burkartii*, in which no direct relationship was found between surface soil moisture and physiological variables estimated at the seasonal level [22]. At the climatic level, we highlight the strong positive relationship between environmental moisture detected in early spring [September] and flower production. Similar results were found by García-Mozo et al. [36] in Mediterranean ecosystems, where periods of high environmental humidity directly influence pollen and fruit production in the genus *Quercus*. Furthermore, we observed high intraspecific variability in fruit production, consistent with the findings of Laouali et al. [37] and Risio et al. [38]. These trends seem to be common in the genus *Prosopis* and tend to be associated with the high genetic variability that can be found at the population level in the species belonging to both sections Strombocarpa and Algarobia of this legume genus [39]. *Prosopis strombulifera* is a woody halophyte shrub adapted to extreme conditions in the Atacama Desert ecosystem. However, current threats with respect to water extraction in this area require further studies on buffer strategies that this species may require in the future under a scenario of decreasing water availability.

## 5. Conclusions

We highlight the importance of ecophysiological variables during the spring period regarding the reproductive aspects of *P. strombulifera*, a halophyte legume that is considered invasive in the deserts of North America and is used as an agroforestry plant in South America. Factors related to water use and fluorescence chlorophyll content during spring can be a powerful tool to determine predictive mechanisms of fruit production in this species in South American ecosystems in order to estimate the carrying capacity for animals or to monitor the reproduction of this species under natural conditions. As it is predominantly invasive in the USA, our results may be relevant regarding the invasive potential of this plant. Therefore, monitoring of spring water and fluorescence parameters, such as SLA, relative water content, and NPQ, plays a key role in the competitive and invasive success of

*P. strombulifera* in arid areas of North America. The ecological dynamics in northern Chile's deserts are groundwater recharge event-driven. Given the current high water extraction in this ecosystem, it is crucial to prioritize the continuous monitoring of ecologically important species such as those belonging to the *Prosopis* genus that are present in this hyperarid ecosystem. Our study has revealed a high relationship between water parameters and reproductive variables. In addition, the recent threat of the climate variability scenario is added, which would considerably affect desert ecosystems. We expect that in the future these areas will become hotter and drier.

**Author Contributions:** F.S.C. and R.C. conceptualized and designed the study. F.S.C., R.C. and E.C. collected, compiled, and analyzed all the data. F.S.C. and R.C. led the writing of the manuscript, and E.B. reviewed and edited all drafts. F.S.C., R.C., E.C. and E.B. read and approved the final manuscript text. All authors have read and agreed to the published version of the manuscript.

**Funding:** This research was funded by the Millennium Nucleus on Applied Historical Ecology for Arid Forests [Aforest] (ANID NCS2022_024).

**Data Availability Statement:** Data from this study have been deposited in the following code repository: https://github.com/carevicunap/DATA-RAW.git (accessed on 14 April 2023).

**Acknowledgments:** We thank C. Chavez, CONAF, and D. Castro for their dedication and contribution to this study in field support.

**Conflicts of Interest:** The authors declare no conflict of interest.

## Abbreviations

Fv/Fm, ratio of variable fluorescence to maximal fluorescence; NPQ, nonphotochemical quenching; Pq, photochemical quenching; RWC, relative water content; RWCa, apoplastic relative water content; SLA, specific leaf area.

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
