# Peer review of "Identifying Plant Physiological and Climatic Drivers in the Woody Shrub Prosopis strombulifera: Effects of Spring Traits on Flower Sprouting and Fruit Production"

_forests, doi:10.3390/f14061167_

Round 1

Reviewer 1 Report

Overall comments:

This paper studies Prosopis strombulifera growing in arid regions of America, examining the relationship between fruit/flower production and leaf ecophysiology. The paper is of high value because of the extensive data collection, including measurements of many ecophysiological parameters in four seasons over three years. However, the data analysis methods should be improved.

              The authors analyzed the three years of measured data separately for each year and concluded that the ecophysiological parameters that showed significant correlations with fruit or flower production in only one (sometimes two) of the three years strongly influenced reproductive status. However, this could be interpreted to mean that, conversely, these ecophysiological parameters were only weakly related to the reproductive status, since they were not significantly correlated in two of the three years. Since annual variation in fruit and flower production was observed, the authors should also perform data analysis on the three years of combined data. In the data analysis, a simple correlation analysis can be performed on the three years of combined data, or a generalized linear mixed model with year as a random effect can be used. These additional data analyses may provide more reliable relationships between ecophysiological traits and fruit and flower production in P. strombulifera.

Specific comments: 

L18 “FP” and “FS”: These abbreviations are not used thereafter and should be deleted.

L26 “A negative correlation was found between the mean of the minimum temperatures of spring and flower sprouting.”: Delete the sentence because it is not discussed in the text. Or discuss it in the text.

L39 “(1)”: References should be cited in []. The same applies to other parts of the document.

L69-71 “We hypothesized that seasonal ecophysiological parameters would strongly affect reproduction parameters.”: Rather, isn't the goal to explore ecophysiological traits that are associated with reproduction parameters? I think that this study does not examine the mechanism by which ecophysiological traits affect reproduction.

L78: Please add the latitude and longitude of the study area.

L80 P. strombulifera”: Add a priod.

L81 “24 ind/ha”: “/ha” should be written as “ha-1”. The same applies where "/" is used for other units (e.g. g/cm2 in line 123).

L81 “P. strombulifera height was measured using a NikonTM Forestry Pro hypsometer.”: Please move the text to a more appropriate position in the manscript. Please match the equipment description to the pressure chamber on line 101.

L86: Please correct the text. Also, please write latitude and longitude instead of UTM.

L92 Figure 1: I recommend that Panel A also be shown as a line chart. Is it soil volumetric moisture content? Please align the horizontal axis of panels A and B so that the correspondence between soil moisture and other meteorological data is clear. This would make it easier for readers like me, who live in the Northern Hemisphere, to understand the correspondence between months and seasons in the figure.

L98 “Digital humidity probes (ECHO)”: Please match the equipment description to the pressure chamber on line 101.

L100: “The water potential (Ψ) pre-dawn” should be “The shoot water potential (Ψ) at predawn”.

L121 “apoplastic relative water content (RWCa)”: Because apoplastic relative water content is not as common as other PV parameters, it requires some explanation.

L130 “fluorimeter”: Unify with "fluorometer.

L140: “pther”?

L141 “from November to January” and L152 “from January to April”: Please add the seasons so that readers living in the northern hemisphere can easily understand.

L174: Correct “analizes” to “analyses”. Did you use Pearson's correlation analysis? It may be better to use Spearman's correlation analysis because the relationship is not always linear (e.g., Carevic et al 2009 Agroforestry Systems).

L178: Please show the results for the seasonal changes in all the parameters. Please also provide a brief description of the trend. Please add when the leaves open and when they fall, or if leaf opening and defoliation will occur throughout the year, as some of the parameters you presented can be related to leaf phenology.

L181 “Seasonal evolution”: Seasonal variation?

L184: Please describe what the number after the plus/minus sign represents (standard error, standard deviation?) as it first appears in the text.

L185 p=0.15: Add spaces.

L198 and Table 2: How about adding new correlations between fruit production and flower sprout data accumulated over three years and ecophysiological parameters in Table 2 or a new Table 3? Many of the parameters described as having significant correlations were shown to be significant for only one year, thus the relationship between the parameters and flower and fruit production is unclear. Since annual variation was observed for fruit production and flower sprouts, this would better clarify the ecophysiological traits that are closely related to fruit production and flower sprouts.

L194 Table 1: Please describe what the number after the plus/minus sign represents (standard error, standard deviation?). Correct the superscripted parenthesis.

L199, L210 RWCa: Unify the "a" in "RWCa" with a subscript. Please check for similar corrections in the manuscript.

L202 Table 2: Please add to the table caption what "sig" and "r" represent. The Greek letters are missing from the names of the three parameters that represent water potential. In the supplementary material, please also include any correlations with ecophysiological parameters measured in other seasons.

L207: For both fruit production and flower sprouts, a significant relationship was found for only one year of the three-year study, i.e., no significant relationship could be seen for two years, which I think is insufficient evidence to consider that RWCa is related to fruit and flower production. As I described above, Spearman's correlation analysis and correlations over the three years should be analyzed to confirm that these relationships are robust. The same applies to the SLA, NPQ, Fv/Fm, etc. discussed below.

              Did the shoots on which you measured the P-V curves contain flowers? I could not understand the mechanism by which turgor maintenance is important at the reproductive stage. Did turgor maintenance contribute to increased drought tolerance and stomatal opening of the leaves, resulting in photosynthesis and thus more flower production, or is turgor maintenance directly required in the floral part during flowering, or is there some other mechanism?

L211: Reference #21 did not study Prosopis and is therefore inappropriate as a citation. Please cite the appropriate reference. Also, please clarify which organ of the plant or the entire plant you are referring to for water flow or the hydraulic conductivity parameters.

L212 “cold cavitation”: What is cold cavitation? If you are referring to cavitation caused by the freeze-thaw cycle that occurs in winter, then of course cavitation will not occur until the temperature drops below freezing. From Figure 1, I believe that the winter cold is mild, and therefore cold cavitation does not occur.

              Regarding the seasonal change in RWCa, you discuss its relevance to water flow. I honestly do not understand what characteristics RWCa indicates, but is it possible that RWCa is caused by changes in leaf morphology? In fact, in Figure 2, the decrease in SLA seems to coincide with the decrease in RWCa.

L229 “shorter photoperiods”: I think the shorter photoperiod is more likely to prevent the development of photoinhibition.

L262 “powerful tool”: A significant relationship for only one year out of three is not a powerful tool.

L279 References: All references with a doi should describe the doi. The format of the doi description should follow the Instructions for Authors.

Author Response

              The authors analyzed the three years of measured data separately for each year and concluded that the ecophysiological parameters that showed significant correlations with fruit or flower production in only one (sometimes two) of the three years strongly influenced reproductive status. However, this could be interpreted to mean that, conversely, these ecophysiological parameters were only weakly related to the reproductive status, since they were not significantly correlated in two of the three years.

Since annual variation in fruit and flower production was observed, the authors should also perform data analysis on the three years of combined data. In the data analysis, a simple correlation analysis can be performed on the three years of combined data, or a generalized linear mixed model with year as a random effect can be used. These additional data analyses may provide more reliable relationships between ecophysiological traits and fruit and flower production in P. strombulifera.

  1. We decided to place the analyzes by separate years for the following reasons:

1) data autocorrelation

2) each year has different climatic variables

3) high variability of production in the genus between years

In relation to the first cause, one of the basic statements to carry out the analysis separated by years, is due to the autocorrelation of data that would exist when applying linear models. Previous articles reviewing the same goals as ours recommend running these analyzes separately for this reason. Thus, in the case of the effect of pollen on fruits (García mozo et al., 2007), climate in fruits (Alejano et al., 2009; Andivia et al., 2012) or climate in water use by Prosopis (Dzikiti et al. ., 2013). In this sense, identifying each year which parameter influences the production of fruits, can be a predictive tool for forest entities in the case of requiring management in these species due to lack of water or other nutrient depending on the parameter (e.g. SLA). If the years are joined, it would not be clear to know which parameter affects flower or fruit production in a given year. However, we believe that the number of variables that were related to reproductive parameters were not scarce, and other authors point out one or two important parameters in this type of relationship.

Secondly, climatic variables in this part of the Atacama Desert vary from year to year, and as concluded or deduced from the manuscript, climatic variables affect the physiological parameters of the species. For example, during years with the presence of the El Niño Southern Oscillation (ENSO) (as may be the case of this year 2023), humidity and rainfall tend to change in relation to the rest of the years, which favors the annual analysis of these variables and their effects in the reproduction.

Climate variables in La Huayca during ENSO years.

Other scientific reason for show the data per separated year, is because Prosopis species does not show fruit production similar between years. This is because these species show intermittent synchronous fruit production (i.e., masting) and this behavior affects various forest ecosystem processes such as forest regeneration and mice-population dynamics. Similarly, are there cases that annual fluctuations in fruit production of Prosopis species affects ecosystem processes or economic activity? It is easy to understand the context of the study if some problems caused by annual fluctuations in fruit production is described (e.g., shortage of fruits for livestock breeding or raising seedlings in a poor-crop year). In other study, we show the intermittent fruit production in Prosopis (Carevic et al., 2021).

Alejano R, Tapias R, Fernández M, Torres E, Alaejos J, Domingo J. Influence of pruning and the climatic conditions on acorn production in holm oak (Quercus ilex L.) dehesas in SW Spain. Ann For Sci. 2008; 65: 209-219.

Andivia E, Carevic F, Fernández M, Alejano R, Vázquez-Piqué J, Tapias R (2012a). Seasonal evolution of water status after outplanting of two provenances of Holm oak nursery seedlings. New Forests 43: 815-824.

Carevic, F.; Alarcón, E.; Villacorta, A.  Is the visual survey method effective for measuring fruit production in Prosopis tree species?. Rangeland Ecol. Manag. 2021, 74:119-124.

Dzikiti, S.; Ntshidi, Z.; Le Maitre, D.C.; Bugan, R.D.; Mazvimavi, D.; Schachtschneider, K.; Pienaar, H.H. Assessing water use by Prosopis invasions and Vachellia karroo trees: Implications for groundwater recovery following alien plant removal in an arid catchment in South Africa. For. Ecol. Manag. 2017, 398, 153–163.

García-Mozo H, Gómez-Casero MT, Dominguez E, Galán C. Influence of pollen emission and weather-related factors on variations in holm-oak (Quercus ilex subsp. ballota) acorn production. Environ Exp Bot. 2007; 61: 35-40.

Specific comments: 

L18 “FP” and “FS”: These abbreviations are not used thereafter and should be deleted.

  1. ok, we deleted these abbreviations

L26 “A negative correlation was found between the mean of the minimum temperatures of spring and flower sprouting.”: Delete the sentence because it is not discussed in the text. Or discuss it in the text.

  1. Now, we discuss briefly this point.

L39 “(1)”: References should be cited in []. The same applies to other parts of the document.

  1. Ok. FIxed

L69-71 “We hypothesized that seasonal ecophysiological parameters would strongly affect reproduction parameters.”: Rather, isn't the goal to explore ecophysiological traits that are associated with reproduction parameters? I think that this study does not examine the mechanism by which ecophysiological traits affect reproduction.

  1. We are not referring to mechanisms, but to which exact parameter influences the reproductive aspects of the species.

L78: Please add the latitude and longitude of the study area.

  1. ok.

L80 P. strombulifera”: Add a priod.

R.ok.

L81 “24 ind/ha”: “/ha” should be written as “ha-1”. The same applies where "/" is used for other units (e.g. g/cm2 in line 123).

  1. We corrected the expression

L81 “P. strombulifera height was measured using a NikonTM Forestry Pro hypsometer.”: Please move the text to a more appropriate position in the manscript. Please match the equipment description to the pressure chamber on line 101.

  1. ok

L86: Please correct the text. Also, please write latitude and longitude instead of UTM.

R.ok

L92 Figure 1: I recommend that Panel A also be shown as a line chart. Is it soil volumetric moisture content? Please align the horizontal axis of panels A and B so that the correspondence between soil moisture and other meteorological data is clear. This would make it easier for readers like me, who live in the Northern Hemisphere, to understand the correspondence between months and seasons in the figure.

  1. We redid both figures for your better comprehension. First, we put the soil data as g water/g soil and modified the scale x axis (we believe that this is the correct form for soil data, as other articles published) and the climate figure now contains arrows for the beginning of the reproduction season.

L98 “Digital humidity probes (ECHO)”: Please match the equipment description to the pressure chamber on line 101.

  1. ok.

L100: “The water potential (Ψ) pre-dawn” should be “The shoot water potential (Ψ) at predawn”.

  1. ok.

L121 “apoplastic relative water content (RWCa)”: Because apoplastic relative water content is not as common as other PV parameters, it requires some explanation.

  1. ok

L130 “fluorimeter”: Unify with "fluorometer.

R.ok

L140: “pther”?

R.ok

L141 “from November to January” and L152 “from January to April”: Please add the seasons so that readers living in the northern hemisphere can easily understand.

R.ok

L174: Correct “analizes” to “analyses”. Did you use Pearson's correlation analysis? It may be better to use Spearman's correlation analysis because the relationship is not always linear (e.g., Carevic et al 2009 Agroforestry Systems).

  1. We use Spearman correlation

L178: Please show the results for the seasonal changes in all the parameters. Please also provide a brief description of the trend. Please add when the leaves open and when they fall, or if leaf opening and defoliation will occur throughout the year, as some of the parameters you presented can be related to leaf phenology.

  1. The objective of showing some parameters in the figures is due to the fact that these are precisely the parameters in which seasonal differences were found (RWCa, apoplastic relative water content, Fv/Fm, ratio of variable fluorescence to maximal fluorescence, NPQ, nonphotochemical quenching and SLA, specific leaf area), therefore they contribute to achieving the objective of the article. These species remain with leaves all year, and leaf senescence is mainly due to water scarcity (Calderón et al., 2015).

Calderón, G.;  Garrido, M.;   Acevedo, E. Prosopis tamarugo Phil.: a native tree from the Atacama Desert. Groundwater table depth thresholds for conservation. Rev. Chil. Hist. Nat. 2015, 88, 18. doi: 10.1186/s40693-015-0048-0

L181 “Seasonal evolution”: Seasonal variation?

R.OK

L184: Please describe what the number after the plus/minus sign represents (standard error, standard deviation?) as it first appears in the text.

  1. OK

L185 p=0.15: Add spaces.

R.OK

L198 and Table 2: How about adding new correlations between fruit production and flower sprout data accumulated over three years and ecophysiological parameters in Table 2 or a new Table 3? Many of the parameters described as having significant correlations were shown to be significant for only one year, thus the relationship between the parameters and flower and fruit production is unclear. Since annual variation was observed for fruit production and flower sprouts, this would better clarify the ecophysiological traits that are closely related to fruit production and flower sprouts.

  1. Thanks for the recommendation. We did this relationship in table 2, so I do not clear what could be the new result

L194 Table 1: Please describe what the number after the plus/minus sign represents (standard error, standard deviation?). Correct the superscripted parenthesis.

R.OK

L199, L210 RWCa: Unify the "a" in "RWCa" with a subscript. Please check for similar corrections in the manuscript.

R.OK

L202 Table 2: Please add to the table caption what "sig" and "r" represent. The Greek letters are missing from the names of the three parameters that represent water potential. In the supplementary material, please also include any correlations with ecophysiological parameters measured in other seasons.

R.OK

L207: For both fruit production and flower sprouts, a significant relationship was found for only one year of the three-year study, i.e., no significant relationship could be seen for two years, which I think is insufficient evidence to consider that RWCa is related to fruit and flower production. As I described above, Spearman's correlation analysis and correlations over the three years should be analyzed to confirm that these relationships are robust. The same applies to the SLA, NPQ, Fv/Fm, etc. discussed below.

  1. We recognize that there was a typing error in table 1. What you are mentioning is the difference in flower production that actually existed in 2017. Due to an involuntary error, we had put in that table that in the case of production of fruits, the differences were in 2018, but as can be seen, high fruit productions were estimated in 2016 and 2018, while in 2017 the lowest production was produced. By mistake, we placed 0.02 in the significance of the year 2018, when the correct data is 0.20. In summary, when you tell us that there were differences in a single year, you are effectively referring to flower sprouts (year 2017) and in the case of fruit production, 2017 was the only year in which differences occurred. What had been correctly entered were the trend letters above each average, which alerted us that we possibly had typing errors in said table with respect to the F and their significance. Regarding table 2, no correlations were detected in only 1 year between the reproductive and physiological parameters, for example for SLA there are significant correlations in 2016 and 2017 (2016: 0.01 r:0.55; 2017; 0.01 r:0.50)

              Did the shoots on which you measured the P-V curves contain flowers? I could not understand the mechanism by which turgor maintenance is important at the reproductive stage. Did turgor maintenance contribute to increased drought tolerance and stomatal opening of the leaves, resulting in photosynthesis and thus more flower production, or is turgor maintenance directly required in the floral part during flowering, or is there some other mechanism?

  1. Water movement is important for opening of the flower, since the process is impaired in rose cut flowers if there is a blockage in the basal stem. In some other species (e.g. cotton and chrysanthemum), the continuum of water in the flower is separated from that in the stem, and the flower can open even when leaves are wilting due to desiccation. Such separation may be highlighting the importance of water relations in flower opening and development in general. Young flowers typically contain high levels of starch. Solute levels increase prior to flower opening through the uptake of sugars from the apoplast and the conversion of polysaccharides (starch, fructan or both) to monosaccharides (fructose and glucose).

L211: Reference #21 did not study Prosopis and is therefore inappropriate as a citation. Please cite the appropriate reference. Also, please clarify which organ of the plant or the entire plant you are referring to for water flow or the hydraulic conductivity parameters.

  1. We corrected the reference

L212 “cold cavitation”: What is cold cavitation? If you are referring to cavitation caused by the freeze-thaw cycle that occurs in winter, then of course cavitation will not occur until the temperature drops below freezing. From Figure 1, I believe that the winter cold is mild, and therefore cold cavitation does not occur.

  1. In La Huayca (área of our study) mínimum absolutes temperatures below zero occurs from April to October. The figure 1 shows mean temperature, obviously during the day from May to August the temperatures reach close to 30 degrees Celsius. A colleague, recently publishes the detail of the temperature in the study area:

https://www.scielo.cl/scielo.php?script=sci_arttext&pid=S0718-34292022000200027&lng=es&nrm=iso&tlng=es

Minimum absolutes temperatures at La huayca.

              Regarding the seasonal change in RWCa, you discuss its relevance to water flow. I honestly do not understand what characteristics RWCa indicates, but is it possible that RWCa is caused by changes in leaf morphology? In fact, in Figure 2, the decrease in SLA seems to coincide with the decrease in RWCa.

  1. Yes, SLA is strongly dependent of RWCa , that’s is the ecological relationship between the both parameters. Curiously, RWCo did not have the same relationship with the reproduction parameters, a fact that I think can be investigated in greater detail in the future, but which may be due torehydration processes in this genre. At physiological level, the plateau effect was interpreted as excess water that is stored in the apoplast, which contributes to the changes in water content and buffers against changes in Ψ. It is likely that this apoplastic water is stored in the intercellular spaces of leaves, which can occupy between 5 and 40% of the total leaf volume. Apoplastic water is the water that resides in xylem vessels and cell walls. The amount of apoplastic water is often small, but it has been frequently reported that apoplastic water has a negligible amount of active solutes when compared with symplast.

L229 “shorter photoperiods”: I think the shorter photoperiod is more likely to prevent the development of photoinhibition.

  1. Could be, but this species are adapted to received 1,200 watts per square meter, and the stomatal closure occurs between 15 and 16 hours in the afternoon.

L262 “powerful tool”: A significant relationship for only one year out of three is not a powerful tool.

  1. No these fact is not so, we found relationships among physiological parameters in more than 1 year in our article, please, if you see the table you can find the significant relationships in different years, this differences are in black colour:

Parameter

[n=25]

Fruit production 2016

Fruit production 2017

Fruit production 2018

Flower sprout 2016

Flower sprout 2017

Flower sprout 2018

sig

r

sig

r

sig

r

sig

r

sig

r

  sig

r

predawn

1.90

0.03

2.26

<0.01

3.08

<0.01

3.65

<0.01

3.22

<0.01

2.95

<0.01

100

2.03

<0.01

2.51

<0.01

2.77

<0.01

3.18

<0.01

3.26

<0.01

2.45

<0.01

     0

0.17

0.11

0.13

0.12

0.32

0.06

1.24

0.03

0.21

0.10

0.18

0.11

RWC0

0.20

0.24

0.15

0.18

0.23

0.09

1.25

0.11

1.55

0.08

1.19

0.13

RWCa

0.15

0.34

0.12

0.12

<0.01

0.72

0.08

0.62

0.03

0.67

0.11

0.49

Fv/Fm

0.16

0.09

0.03

0.59

0.09

0.23

0.10

0.18

0.11

0.20

0.01

0.60

NPQ

0.02

-0.54

0.13

-0.27

0.01

-0.61

0.12

-0.38

<0.01

-0.63

0.18

-0.12

Pq

4.32

<0.01

4.98

<0.01

5.15

<0.01

4.89

<0.01

4.96

<0.01

4.55

<0.01

    SPAD

3.61

<0.01

3.78

<0.01

4.33

<0.01

4.93

<0.01

3.21

<0.01

2.88

<0.01

SLA

0.01

0.55

0.01

0.50

0.10

0.40

0.21

0.15

<0.01

0.70

0.20

0.10

L279 References: All references with a doi should describe the doi. The format of the doi description should follow the Instructions for Authors.

R.ok,

Reviewer 2 Report

Over all I found the manuscript cogent and well-organized.

line 195-199 and then flowing to the abstract:  Since you cannot have fruit without the flower, the results and the relationship showing the positive correlation is rather obvious and more of botanical mechanics than otherwise, unless you show a comparative flower to fruit success ratio.  that the env cues for success in flowering then impact yield in fruit.....expected, so emphasize (to great degree)  the inevitability of the relation and focus on the flower period.  If fecundity was impacted...then linked with fruit development % of the flowers in place....you could make more of the fruiting data.

Author Response

Over all I found the manuscript cogent and well-organized.

line 195-199 and then flowing to the abstract:  Since you cannot have fruit without the flower, the results and the relationship showing the positive correlation is rather obvious and more of botanical mechanics than otherwise, unless you show a comparative flower to fruit success ratio.  that the env cues for success in flowering then impact yield in fruit.....expected, so emphasize (to great degree)  the inevitability of the relation and focus on the flower period.  If fecundity was impacted...then linked with fruit development % of the flowers in place....you could make more of the fruiting data.

  1. Thanks for the comment. We focus precisely on seed production, but at a level of how physiological variables influence them each year. If the comment is from the point of view of a more complete analysis of fruit production, we would need more years to detect masting, good producers or bad producers. Our objective is based on knowing each year which variable affects the production of flowers and fruits as a decision-making tool, since the climate is different each year in this area, from an environmental point of view.

Round 2

Reviewer 1 Report

There were corrections to my previous specific comments, but almost no correction to the crucial overall major comment.

The year-by-year correlation analysis in Table 2 showed a significant positive correlation between fruit production and RWCa in 2018. However, no significant correlation was found in 2016, when fruit production was as high as in 2018. Also, despite the high significant correlation between fruit production and flower production, no significant correlation was found between RWCa and flower production in 2018, rather a significant correlation was only found between flower production and RWC in 2017, when flower and fruit production was low. Thus, it is unreasonable to claim that "RWCa was strongly correlated with fruit production and flower parameters (in line 273)" because no significant correlation was found in two of the three years, and the year in which a significant correlation was found was not consistent with fruit and flower production.

To eliminate these data analysis vulnerabilities, in my previous comment, I tried to recommend adding correlation analyses summarizing three years of data in Table 2, but I guess my suggestion was not clear to you. In Table 2, it was fine to show the correlation of fruit production or flower sprout in each year with each physiological parameter measured in the spring for each year as you presented. But I suggested that the correlations of fruit production or flower sprout in each year with each physiological parameter measured in the spring should also be shown when the data set is combined into one data set for three years. In other words, please also show the correlations for 75 data points (25 individuals x 3 years = 75 data points) that include a high and a low fruit and flower production year. This would better clarify the physiological traits that are closely related to fruit production and flower buds. I think this type of analysis would not have the autocorrelation problem you have in mind. In Garcia-Mozo et al 2007 and Alejano et al 2008, which you described in your response, they actually combined the data from each year into one and did a correlation analysis, as I recommended. Although you cited the different climates and different fruit and flower production from year to year as the reason for analyzing the different years separately, I think it is rather important under these conditions to do a correlation analysis with three years of data as one data set. In some cases, physiological changes in response to production should become more apparent for the analysis.

In the discussion, you mentioned turgor maintenance during the reproductive stage in the context of winter cold cavitation as the reason for correlating RWCa with flower and fruit production. Why didn't you calculate the turgor potential from the measured predawn water potential and the osmotic potential at the turgor loss point? In your discussion, you mention the maintenance of turgor during the reproductive season to account for cold winter cavitation as a reason for the correlation between RWCa and flower and fruit production. Why did you not calculate turgor pressure from pre-dawn water potential and osmotic potential at the point of turgor loss? Also, the daily mean temperature data you presented did not provide evidence for cold cavitation because you did not show whether the daily minimum temperature was below zero or not in winter. In addition, you state on line 254 that "RWCa is closely related to water flow in Prosopis, specifically to hydraulic conductivity [K] parameters [21]. ", but in paper #21 they measured neither RWCa nor in situ hydraulic conductivity, they only measured theoretical hydraulic conductivity from vessel dimensional parameters, and they did not examine the relationship between RWCa and hydraulic conductivity. In addition, they refused to provide data on predawn water potential or other water relations parameters. Under these circumstances, it is impossible to find a relationship that spring RWCa has a strong influence on spring flower and fruit production through recovery from winter cold cavitation.

Regarding Table 2, last time I commented that you should add the meaning of "sig" and "r", and you have explained that "r" is "Spearman correlations" and "sig" is "statistical significance". Are these correct? First, shouldn't "r" be "Spearman correlation coefficients"? Also, in general, “statistical significance” is not a statistic value that appears as a number, but a term used to say that an observation is "unlikely to have occurred under the null hypothesis of a statistical test". To determine "statistical significance", a p-value or probability value (a number between zero and one) is usually used for a criterion (often p < 0.05). However, the "sig" you described has a value greater than 1. Please clearly state what your "sig" statistic is.

Author Response

The year-by-year correlation analysis in Table 2 showed a significant positive correlation between fruit production and RWCa in 2018. However, no significant correlation was found in 2016, when fruit production was as high as in 2018. Also, despite the high significant correlation between fruit production and flower production, no significant correlation was found between RWCa and flower production in 2018, rather a significant correlation was only found between flower production and RWC in 2017, when flower and fruit production was low. Thus, it is unreasonable to claim that "RWCa was strongly correlated with fruit production and flower parameters (in line 273)" because no significant correlation was found in two of the three years, and the year in which a significant correlation was found was not consistent with fruit and flower production.To eliminate these data analysis vulnerabilities, in my previous comment, I tried to recommend adding correlation analyses summarizing three years of data in Table 2, but I guess my suggestion was not clear to you. In Table 2, it was fine to show the correlation of fruit production or flower sprout in each year with each physiological parameter measured in the spring for each year as you presented. But I suggested that the correlations of fruit production or flower sprout in each year with each physiological parameter measured in the spring should also be shown when the data set is combined into one data set for three years. In other words, please also show the correlations for 75 data points (25 individuals x 3 years = 75 data points) that include a high and a low fruit and flower production year. This would better clarify the physiological traits that are closely related to fruit production and flower buds. I think this type of analysis would not have the autocorrelation problem you have in mind. In Garcia-Mozo et al 2007 and Alejano et al 2008, which you described in your response, they actually combined the data from each year into one and did a correlation analysis, as I recommended. Although you cited the different climates and different fruit and flower production from year to year as the reason for analyzing the different years separately, I think it is rather important under these conditions to do a correlation analysis with three years of data as one data set. In some cases, physiological changes in response to production should become more apparent for the analysis.

  1. We believe that you are correct in grouping the analyzes into 75 observations. In a first review, it was not very clear to us what you wanted done, but in this second opportunity, we have grouped the data in the search for correlations with the 75 observations. Although this new analysis indicates the relationships between the physiological parameters and the production of flowers and the number of estimated fruits, we believe that the relationships between RWCa were consistent with its direct effect on both reproductive variables of the species.

In the discussion, you mentioned turgor maintenance during the reproductive stage in the context of winter cold cavitation as the reason for correlating RWCa with flower and fruit production. Why didn't you calculate the turgor potential from the measured predawn water potential and the osmotic potential at the turgor loss point? In your discussion, you mention the maintenance of turgor during the reproductive season to account for cold winter cavitation as a reason for the correlation between RWCa and flower and fruit production. Why did you not calculate turgor pressure from pre-dawn water potential and osmotic potential at the point of turgor loss?

  1. Those parameters were estimated and are reflected in lines 147 to 150: After plotting the pressure-volume curves and as no over-saturation points were detected on any of them, the following parameters were determined: osmotic potential at full turgor (Yp 100), osmotic potential at the turgor loss point (Yp0), relative water content at the turgor loss point (RWC0), and apoplastic relative water content (RWCa). What happens is that in these parameters we did not find statistical differences during the different dates in which they were estimated and this is how we clarified it in the first two lines of the results section, since there were only four estimates that did reflect seasonal differences to discuss them.

Also, the daily mean temperature data you presented did not provide evidence for cold cavitation because you did not show whether the daily minimum temperature was below zero or not in winter. In addition, you state on line 254 that "RWCa is closely related to water flow in Prosopis, specifically to hydraulic conductivity [K] parameters [21]. ", but in paper #21 they measured neither RWCa nor in situ hydraulic conductivity, they only measured theoretical hydraulic conductivity from vessel dimensional parameters, and they did not examine the relationship between RWCa and hydraulic conductivity. In addition, they refused to provide data on predawn water potential or other water relations parameters. Under these circumstances, it is impossible to find a relationship that spring RWCa has a strong influence on spring flower and fruit production through recovery from winter cold cavitation.

  1. Figure 2 has been modified. The record of the absolute minimum temperatures for each month has been added so that you can visualize that in the winter months (June, July and August) temperatures below zero degrees Celsius are recorded here in the area of our study. In previous studies carried out by our research team, we have determined that there is an effect of winter frosts on physiological parameters. For example, in citation 22, we found a statistically significant decrease in hydraulic conductivity as a result of frost, so it is not surprising that the effect of these temperatures affects RWCa.. Let me tell you that in another article that we carried out (cite 4), we also found an effect of winter temperatures on RWC, the difference was that this study was carried out on another species of Prosopis (4). In this way, we verified that there is a close dependence on the water flow within the species that, as the temperature decreases, manages to generate degrees of cavitation in these trees, which could affect reproductive parameters (table 2).

Regarding Table 2, last time I commented that you should add the meaning of "sig" and "r", and you have explained that "r" is "Spearman correlations" and "sig" is "statistical significance". Are these correct? First, shouldn't "r" be "Spearman correlation coefficients"? Also, in general, “statistical significance” is not a statistic value that appears as a number, but a term used to say that an observation is "unlikely to have occurred under the null hypothesis of a statistical test". To determine "statistical significance", a p-value or probability value (a number between zero and one) is usually used for a criterion (often p < 0.05). However, the "sig" you described has a value greater than 1. Please clearly state what your "sig" statistic is.

  1. The table has been corrected based on your observations. You are absolutely right, the researcher who carried out the statistical part has confirmed the same thing that you mentioned to me, the "sig" value corresponds to p value. Previously, the numbers that were included as "sig" and that were greater than 1 corresponded to other values (F values) described by the statistical software used.
